# Intraoperative 3D quantitative magnetic resonance imaging in paediatric brain tumour surgery

Per Nyman[1,2,3]*, Rafael Turczynski Holmgren[4,5], Emma Nordh[3,5], Anna Ljusberg[1,2,6], Oscar Snödahl[1,2,7], Frederik Testud[8], Ida Blystad[1,2,7], Peter Lundberg[1,2,6,7], Anders Tisell[1,2,6]

**1** Center for Medical Image Science and Visualization' (CMIV), Linköping University, Linköping, Sweden, **2** Department of Medical and Health Sciences, Linköping University, Linköping, Sweden, **3** Crown Princess Victoria Children's Hospital, Region Östergötland, Linköping, Sweden, **4** Clinical Department of Neurosurgery, Region Östergötland, Linköping, Sweden, **5** Department of Biomedical and Clinical Sciences, Linköping University, Linköping, Sweden, **6** Clinical Department of Medical Radiation Physics, Region Östergötland, Linköping, Sweden, **7** Clinical Department of Radiology, Region Östergötland, Linköping, Sweden, **8** Siemens Healthcare AB, Malmö, Sweden and Medical Radiation Physics, Clinical Sciences Lund, Lund University, Lund, Sweden

* per.nyman@liu.se

## Abstract

### Purpose

- To investigate if $R_1$ and $R_2$ can reliably be measured using 3D quantitative MRI in an intraoperative setting when paediatric brain tumour surgery is performed.
- To determine whether $B_1^+$ inhomogeneities affect $R_1$ and $R_2$ measurements in normal-appearing white matter and the thalamus, respectively, and how $R_1$ and $R_2$ measurements are affected by different coils.
- To assess how the relaxation parameters of brain tissue are affected by the intraoperative setting.

### Methods

The accuracy of $R_1$ and $R_2$, the effect of $B_1^+$-field inhomogeneity and how the flex coil position affected $R_1$ and $R_2$ were evaluated, both pre- and intraoperatively during surgery. Ten patients were recruited, six girls and four boys aged 2–15 years, with varying tumour entities, all referred to surgery with intraoperative MR. The patients were scanned using a head coil preoperatively and flex coils intraoperatively. Control experiments were performed on phantoms in various positions, equivalent to the patient positions. ROIs (Regions of Interest) were positioned in areas representing normal-appearing matter. Relaxation rates $R_1$ and $R_2$ were calculated from 3D-quantification using an interleaved Look-Locker acquisition sequence with $T_2$ preparation pulse (3D-QALAS) data.

**Data availability statement:** The phantom data that support the findings of this study are available on the zenodo repository, https://zenodo.org (10.5281/zenodo.17716583). The patient data are not publicly available due to privacy or ethical restrictions.

**Funding:** This work was supported by the Swedish Childhood Cancer Fund https://www.barncancerfonden.se (BCF-MT2019-0027, PR2020-0071) (PL), The Joanna Cocozza Foundation for Children's Medical Research https://liu.se/en/article/joanna-cocozzas-stiftelse. Grants from the Swedish state under the agreement between the Swedish government and the county councils (the ALF-agreement Östergötland) https://www.researchweb.org/is/regionostergotland RÖ-974566 (PL) , RÖ-994768 (PL), RÖ-1012372 (IB), RÖ-990893 (IB) Research Support in the Field of Cancer at Linköping University/County Council of Östergötland (PL) Mary Beve's foundation (PL, PN) http://www.marybeve.se. IB holds a grant as an associated clinical fellow at the Wallenberg Centre for Molecular Medicine (WCMM) https://liu.se/en/research/wcmm. The funder provided support in the form of salaries for authors [PN, AL, IB, PL], but did not have any additional role in the study design, data collection and analysis, decision to publish, or preparation of the manuscript. The specific roles of these authors are articulated in the 'author contributions' section.

**Competing interests:** The author Frederik Testud of this manuscript have the following competing interests: Employee of Siemens Healthineers AB, Sweden.

## Results

There was a significant increase of $R_2$ in the intraoperative setting compared to the preoperative 3D-QALAS measurements. In contrast to the patient examinations, control experiments using relaxation phantoms did not demonstrate similar differences.

## Conclusion

Relaxometry is feasible in the intraoperative setting. The detected differences between the quantitative $R_2$ values in tissue pre- and intraoperatively seem to be explained by the physiological conditions characterising the surgical situation.

## Introduction

Each year, approximately 340 children are affected by cancer in Sweden [1]. The annual frequency is approximately the same in the rest of Europe [2,3]. Tumours in the central nervous system are the second most common tumour in children, comprising approximately 100 cases annually in Sweden with an age-standardized incidence of around 40 per 1,000,000 individuals, and represent the tumour group in children and adolescents with the highest mortality and morbidity [4].

When a child is diagnosed with a brain tumour, neurosurgery is the primary treatment in the majority of cases. The possibility of intraoperative MR diagnostics during the neurosurgical intervention creates unique opportunities for instant assessment of resection extent, thus improving conditions for brain tumour surgery. Performing intraoperative MRI enhances the possibilities for radical and safe resection but requires time optimisation [5–7].

The use of intraoperative MRI usually leads to further tumour resection. There are studies where intraoperative MRI confirms complete resection in only about half of tumour operations considered, where surgical resection was extended in a fifth of the intended cases [8]. Previous work has also shown that the last intraoperative MRI examination is equivalent to postoperative MRI examinations (acquired within 72 h), with respect to the identification of residual tumour [9]; thus, the postoperative examinations are not necessarily required.

Overall, this suggests that intraoperative MRI offers benefits, particularly for children, as the examination can be performed under the same anaesthesia.

However, there are also difficulties with intraoperative MRI since flex coils need to be used, which gives lower SNR, and the head is placed off-centre, which causes larger $B_0$ and $B_1^+$ inhomogeneities.

Conventional MRI used in the clinical routine, is associated with a subjective assessment of signal intensities that vary with different tissues, different coils, sequences, or type of scanner. Quantitative MRI (qMRI) makes it possible to quantify tissue-specific parameters such as relaxation rates, $R_1$ and $R_2$ [10,11], rather than rely on subjective evaluation of magnetic resonance signals. $R_1$ and $R_2$ values make it possible to create synthetic MR images, perform tissue classifications such as identifying grey or white matter, and to calculate myelin concentration [12,13].

This technique has been applied to some adult cohorts for analyses of brain tumours [12], but for the paediatric population, few studies have been performed [14]. Xu *et al.* measured $T_1$ and $T_2$ relaxation values in preoperative screening of brain development in children with congenital heart disease [15]. These values were then considered as a quantitative assessment related to, among other things, myelin development in these children. Relaxation values from different brain regions were also compared with a healthy control group of infants and the findings suggested that the technique is useful and feasible as a reference for assessing brain development for these children [15].

Concerning paediatric brain tumours, there is little to no knowledge about the use of qMRI, and, to our knowledge, none in the perioperative setting during paediatric brain tumour surgery.

The primary aim of this work was therefore to explore whether $R_1$ and $R_2$ can be measured reliably using 3D quantitative MRI (3D QALAS) in an intraoperative setting when paediatric brain tumour surgery is performed. A second aim was to determine if $B_0$ and $B_1^+$ inhomogeneities affect $R_1$ and $R_2$ measurements in normal-appearing white matter (NAWM) and the thalamus. Additionally, it was examined whether the $R_1$ and $R_2$ measurements were affected by different receiver coils. A third aim was to assess if the relaxation parameters of brain tissue are affected by the intraoperative setting. Accordingly the main hypothesis was that the measured $R_1$ and $R_2$ are not affected by the intraoperative setting.

## Methods

### Patients

Ten patients, six girls and four boys aged 2−15 years, with varying tumour entities and referred for surgery with intraoperative MR, were included in this study from March 2022 to June 2024 (Table 1). The study was approved by the *Swedish Ethical Review Authority* (Dnr 2021−00889, P. Lundberg) in accordance with the Declaration of Helsinki. Patients were recruited prospectively, and patients and guardians received oral and written information and further radiological sequences were subject to written consent before the planned examination.

**Table 1. Inclusion of pediatric patients (see footnote for abbreviations).**

| Age[1] | Sex | Duration[2] | Position[3] | Medication Preoperative [4] | Medication Intraoperative [4] | Diagnosis |
|---|---|---|---|---|---|---|
| 5 | M | 0 | prone | P | P | Pilocytic Astrocytoma WHO grade 1 |
| 10 | F | 20 | supine | No drugs (awake) | P, M | Anaplastic Astrocytoma WHO grade 3 |
| 15 | F | 1 | supine | No drugs (awake) | S, P, M | Pilomyxoitt Astrocytoma, FGRF1-mutation, WHO grade 1–2 |
| 9 | F | 1 | supine | No drugs (awake) | S, P, M | Myxoid glioneuronal tumour, PDGFRA-mutation, WHO grade 1 |
| 2 | F | 0 | supine | S, $N_2O$ (during induction), P | S, $N_2O$ (during induction), P, M | Diffuse astrocytoma, MYB/MYBL1 altered subtype, WHO grade 1 |
| 6 | M | 0 | prone | S, P | S, P, M | Medulloblastoma, molecular subgroup 3/4, WHO grade 4 |
| 9 | F | 1 | supine | No drugs (awake) | P, M | Pilocytic Astrocytoma, BRAF V600E mutation, WHO grade 1 |
| 4 | M | 23 | supine | P | P, M | Diffuse astrocytoma, MYB- MYBL1 altered, WHO grade 1 |
| 4 | F | 8 | supine | P | S, P | Embryonal tumour with multilayered rosettes (ETMR), WHO grade 4 |
| 4 | M | 9 | prone | S (during brief induction), $N_2O$ (during induction), P | S, $N_2O$ (during induction), P, M | Diffuse astrocytoma, MYB- MYBL1 altered, WHO grade 1 |

[1]Age at diagnosis (years); [2]Duration in days between pre- and intraoperative MRI. [3]Intraoperative MR patient position. [4]Medication during pre- and intraoperative MRI. S: Sevofluran; P, Propofol; $N_2O$, nitrogen oxide; M, Mannitol. The following drugs were also used in many cases; Adrenalin, Albumin, Atrakurium, Atropin, Betametason, Catapresan, Efedrin, Esmeron, Fentanyl, Ketamin, Ketanest, Morphine, Remifentanil.

## Magnetic resonance

**Data acquisition.** Data acquisition was performed on a 3 T MAGNETOM Skyra 70 cm bore MR scanner (Siemens Healthineers AG, Forchheim, Germany). All MR examinations were performed on the stationary MR scanner next to the operating theatre. The system was set up for both diagnostic imaging with a regular patient table and standard head coils, and for intraoperative examinations where the patient's head is fixed in a headrest and flex coils are used. Figs 1 and 2 present a schematic overview of all measurements, and data analysis (in Fig 1) as well as the phantom versus patient setup and placement in the magnet bore (Fig 2). All experiments were conducted with the scanner's baseline shim settings ("tune-up").

A 3D-QALAS research sequence was implemented containing interleaved Look-Locker acquisitions series for fast quantification of the longitudinal $R_1$ and the transverse $R_2$ relaxation rates were used here. The 3D-QALAS acquisitions [16,17] are based on a low flip angle and short $t_{TR}$ readout; two BIREF-1 refocusing pulses [18] for the $T_2$-preparation and a non-slice-selective 180° 3D adiabatic inversion pulse were used. Protocols were used both preoperatively (1.2x1.2x1.2 mm³) and intraoperatively (1.3x1.3x1.3 mm³). See S1 Data for complete protocol parameters. The total acquisition time was 6 min in both cases.

The 3D-QALAS research sequence and a separate $B_1^+$-mapping sequence with a slice-selective preconditioning radiofrequency pulse [19] were added to both the conventional clinical pre- and the intraoperative MR protocol. Moreover, both pre- and post-gadolinium contrast agent administration was used, when possible, in the clinical pipeline. See Fig 3 for a schematic representation of the measurements in relation to the pulse sequence time-steps.

**Patient MR measurements.** Preoperatively, the patients were scanned using either a 20- or a 64-channel head/neck coil, depending on the patient's head circumference and necessary additional equipment such as a respiratory tube; the 20-channel head/neck coil being more spacious than the 64-channel head/neck coil.

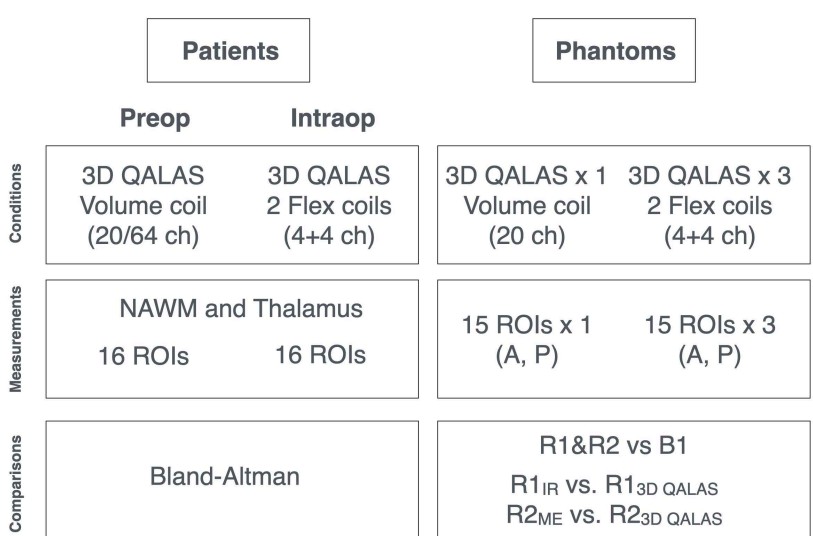

**Fig 1. Schematic overview of the MR protocols and measurements for both patients and phantoms.** Patients: (left) A head coil was used preoperatively, whereas a combination of two flex coils was used for the intraoperative setting. For patients, up to 16 ROIs were placed in 3D-QALAS images in each patient, in normal-appearing white matter (NAWM) and the thalamus. Data were compared using Bland-Altman plots. Phantoms: (right) Overview of the control experiments using four different homogenous relaxometric spherical phantoms. Measurements were performed using both a head array coil and the combination of two flex coils, replicating the measurements performed on patients. The positions of the four different phantoms (and flex coils) were moved into three different spatial positions (A/P direction) in the bore of the magnet, resulting in a total of 60 different ROIs in the phantoms. Fifteen ROIs were then placed in the A/P direction of the images, at varying distances from the centre of the phantom (and isocentre). Data were compared using both correlation and Bland-Altman plots, including a comparison of 3D-QALAS with gold standard pulse sequences (IR and ME).

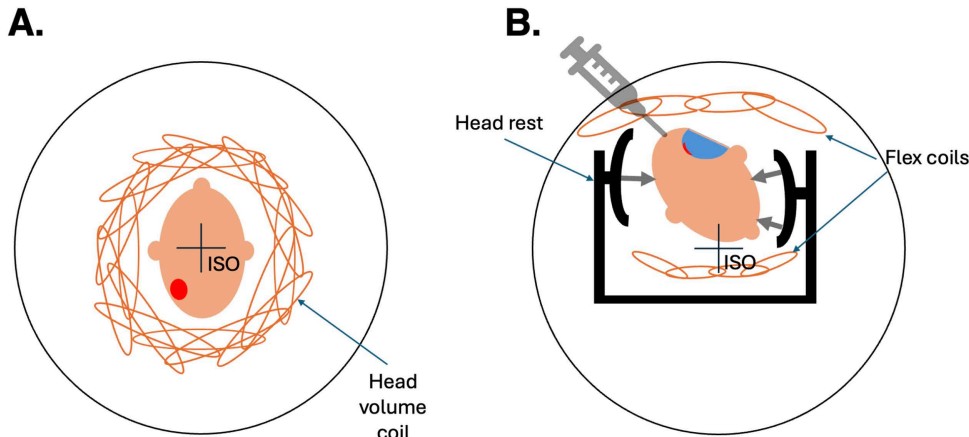

**Fig 2. Schematic overview of the positioning of the patient. (A)** Patient in the preoperative, and **(B)** in the intraoperative settings. The outer circle illustrates the bore of the magnet. Note that different MR detection coils were used (conventional head array coil versus a combination of two flex coils). In addition, the patient positions with comparison to the isocentre of the magnet were very different in the preoperative versus the intraoperative settings due to the spatial requirements of the headrest, and due to the patient-specific determined rotation and height of the patient's head. The headrest was used to fix the patient's head in the appropriate position. Red: Tumour; Blue: Tumour cavity. ISO: Isocentre of the MR scanner.

In the intraoperative setting, two four-channel flex coils were used instead. The patients were placed in a Doro Lucent headrest (Black Forest Medical Group, Freiburg, Germany), and the flex coils were used, as is shown in Fig 2. The patients were anaesthetised, and osmolyte mannitol (Fresenius Kabi AB, Uppsala Sweden) was used during surgery (beginning approximately at the start of surgery) in eight out of ten cases in this study, to reduce intracranial pressure associated with cerebral oedema.

The patients underwent MR examinations before surgery according to the clinical protocol including 3D $T_1$ weighted pre- and post-contrast agent, 3D $T_2$ FLAIR, $T_2$ weighted axial and diffusion weighted images (DWI). In addition, a 3D-QALAS volume was acquired both before and after the administration of gadoterate meglumine (279 mg/mL Dotarem; Guerbet, Villepinte, France) a gadolinium contrast agent ($Gd^{3+}$ CA), as well as $B_1^+$ mapping (performed once).

During surgery, the intraoperative examination was performed according to clinical protocol and in addition, 3D-QALAS was acquired pre- and post-contrast. The patients were examined in either the prone or supine positions, depending on the surgical setting, cf. Table 1.

In two cases, an MRI with gadolinium contrast agent administration was performed at the local hospital and complementary preoperative MRI examinations, including research sequences, was therefore performed without contrast agent.

The research sequences were also performed in four cases only after administration of gadolinium contrast agent, either pre- or intraoperatively. This, together with the loss of ROIs in areas corresponding to pathological tissue, resulted in a total of 2*93 ROIs before (six patients), and 2*117 ROIs (eight patients) after contrast agent administration.

**Phantom MR measurements.** For *in vitro* validation and quality control, a set of four custom-made FUNSTAR phantoms with a diameter of 18 cm (manufactured by Gold Standard Phantoms, Sheffield, United Kingdom) containing 10 mM, 15 mM, 20 mM and 30 mM manganese chloride ($MnCl_2$) were used to obtain different relaxations of the aqueous solution, covering a range of $R_1$ from 1 to 2.5 s$^{-1}$, and $R_2$ from 7 to 22 s$^{-1}$.

The phantoms made it possible to determine how spatial $B_0$ and $B_1^+$ variation affected $R_1$ and $R_2$ values. Two experiments were performed with the four phantoms, to evaluate $R_1$ and $R_2$ accuracy and the effects of $B_0$ and $B_1^+$ -field inhomogeneities. In all phantom experiments the relaxation values were determined at 15 different positions within the phantoms bore.

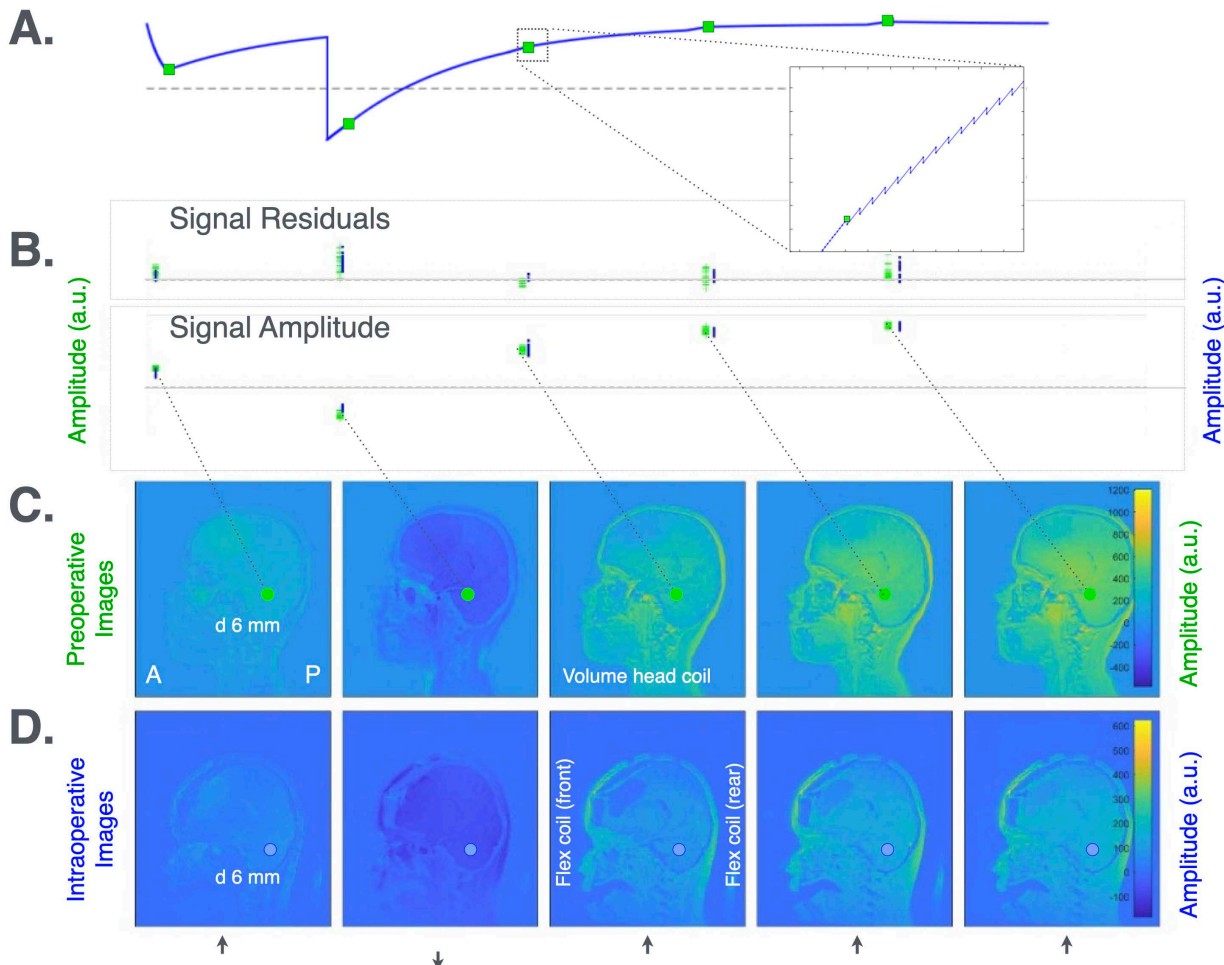

**Fig 3. Example of 3D-QALAS patient measurements. (A)** 3D-QALAS pulse sequence signal pathway. The 3D-QALAS sequence consists of five so called 'dynamics' (see top panel where the five points for data sampling are marked with green squares). Each dynamic consists of a preparation phase and an acquisition phase. In the first dynamic, a T2 preparation pulse is used to make the first volume T2 weighted. The second dynamic starts with a 180 degree-inversion pulse to encode R1 relaxation. In all dynamics, the acquisitions are read with a 4-degree flash readout. **(B)** Signal intensities. Residuals on top and the signal amplitudes in the preop- (green) and intraop situations (blue). **(C)**. Preoperative images. Sagittal images of a patient at different time points in the pulse sequence. (6 mm diameter ROI, containing ca. 14 voxels). The dynamic scale represents signal intensities. **(D)** Intraoperative images. Corresponding images, but in the intraoperative situation. (6 mm diameter ROI, same anatomical location as in (C), containing ca. 11 voxels). See Fig 4 for examples of ROIs in axial slices. The dynamic scale represents signal intensities.

**Accuracy.** Accuracy was assessed in the preoperative setting using the head coil. In addition, to the acquisition of 3D-QALAS, an inversion recovery pulse sequence (IR) and a single slice multi-spin echo sequence (ME) was also used as the gold standard for the $R_1$ and $R_2$ measurements (s) see Supplement data for sequence details.

**$B_1$ effects.** Phantom experiments for determining $B_1^+$ effects were set up using flex coils and positioning the FUNSTAR phantoms in three different off-centre positions using the Doro headrest (see S1 and S2 Figs). 3D-QALAS was acquired using the intraoperative protocol and $B_1^+$ was mapped using the mentioned slice-selective preconditioning radiofrequency pulse [19] to determine if there was any observable $B_1^+$ inhomogeneity. The measured $B_1^+$ factor corresponds to how much the nominal flip angle (FA) deviates from the actual FA.

## Data analysis

**Patient.** Up to 16 standardised regions-of-interest (ROIs), each with a diameter of 6 mm, were placed in synthetic images calculated from the 3D-QALAS volume of the patients for paired analysis of $R_1$ and $R_2$. The ROIs were placed by an experienced (15+years) neuroradiologist (IB) on the axial plane in NAWM and bilaterally in the thalamus, in both pre- and intraoperative images, as well as before and after the injection of gadolinium containing contrast agent (seven on each side and in the corpus callosum genu and splenium). The bilateral ROIs were placed in the following structures (if no pathology): the frontal WM (White Matter), parietal WM, the occipital WM, the peritrigonal WM, lateral to the temporal horn, the cerebellar peduncles, and the thalamus bilaterally. Example ROI positioning is shown in Fig 4. R1 and R2 values in the same locations were compared before and after the contrast agent injection, in both the pre- and intraoperative situations.

**Phantom.** Circular ROIs with a diameter of 6 mm were placed in the 3D-QALAS volume. One ROI was placed in the phantom centre, and 14 additional ROIs were positioned in steps of 10 mm from the phantom centre (anterior, posterior). ROI positioning is shown in S3 Fig. Reference inversion recovery (IR) and multi-echo (ME) images were also acquired and similarly analysed.

**Calculation of relaxation parameters.** Relaxation rates $R_1$ and $R_2$ were calculated from the 3D-QALAS data using an in-house MATLAB script (version R2024a, MathWorks inc., Natick, Massachusetts, United States). $B_1^+$ maps were measured directly on the MR scanner. Gold standard inversion recovery and multiecho were used determine $R_1$ (based on the method presented in [20]) and $R_2$ respectively, using separate MATLAB scripts.

## Statistical analysis

The difference between calculated intraoperative and preoperative $R_1$ and $R_2$ was evaluated using Bland-Altman plots in combination with a paired double-sided Student's t-test. P-values ≤0.05 were considered significant.

## Results

### Patient examinations

The intraoperatively and preoperatively determined $R_1$ and $R_2$ were compared, as were the differences between $B_1^+$ intra-operatively and preoperatively (see Fig 5). Bland-Altman analysis was also performed (Fig 5, panels B and C).

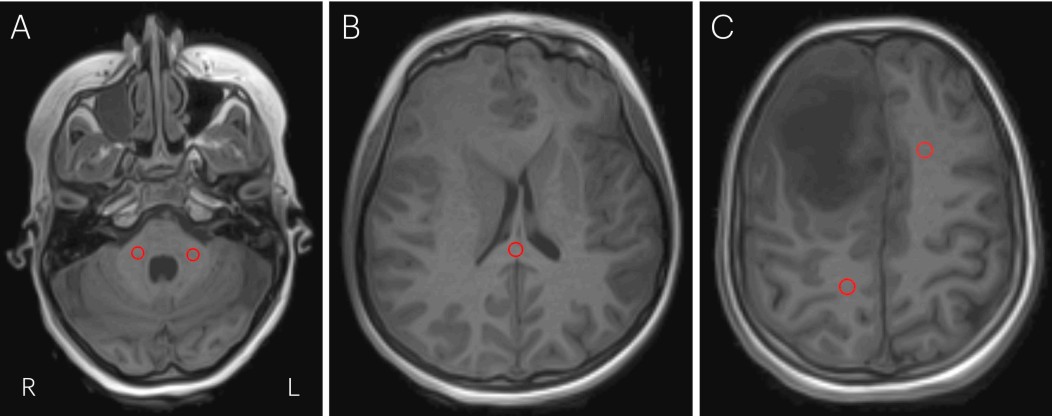

**Fig 4. Typical examples of ROI-placements in patient.** Examples of the placement of ROIs in three different preoperative slices of a patient. Peduncles (LEFT), corpus callosum splenium (MIDDLE), frontal left and parietal right (RIGHT).

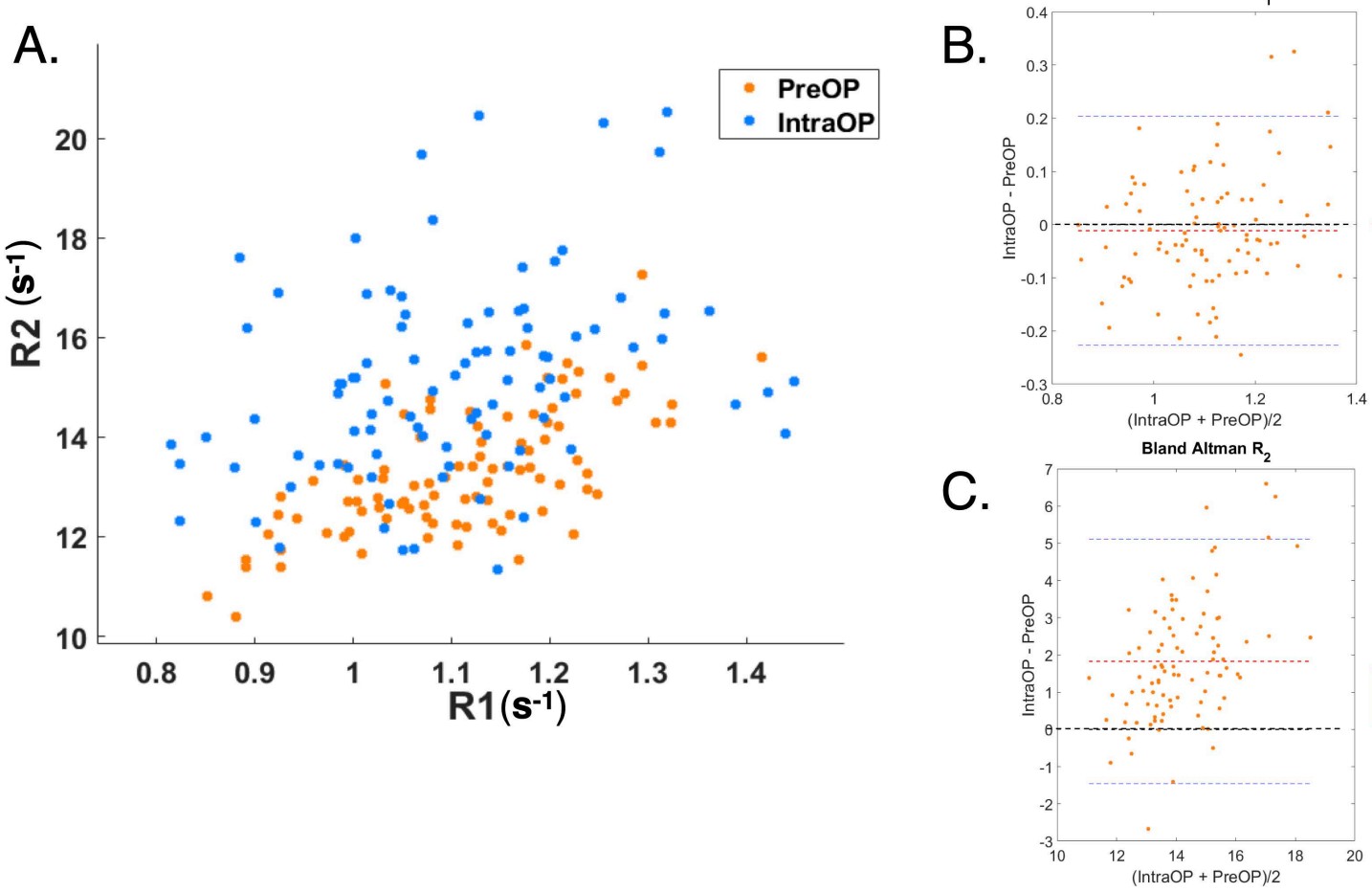

**Fig 5. Correlation between $R_1$ and $R_2$ in patients. (A)** Samples of normal-appearing matter in preoperative vs intraoperative settings. The same structure should have identical values in both settings, everything else being equal. However, a difference was observed between the pre- and intraoperative R2 clusters. In contrast, this was not obvious for R1. Cf. the Discussion for a comprehensive analysis. **(B and C)** Bland-Altman plots of the difference between determined R1 (B) and R2 (C) between the intraoperative and preoperative examinations. The difference was insignificant for R1, but not for R2. An R2 difference of about 2 s-1 was observed.

*No gadolinium contrast agent:* The $R_1$-difference between the intraoperative and preoperative settings was $-0.01 \pm 0.01$ (not significant) and the R2-difference was $1.8 \pm 0.17$ s-1 (**p<0.001**). The $B_1^+$-difference was $-0.041 \pm 0.006$ s-1 (**p<0.001**).

*With gadolinium contrast agent:* The $R_1$-difference between the intraoperative and preoperative settings was $-0.019 \pm 0.021$ (not significant) and the $R_2$-difference was $1.9 \pm 0.18$ s-1 (**p<0.001**). The $B_1^+$-difference was $-0.047 \pm 0.006$ s-1 (**p<0.001**).

In other words, $R_2$ was significantly higher in the intraoperative compared to preoperative examinations, both with and without contrast agent. Using contrast agent, $R_1$ did not differ significantly between the intraoperative and preoperative examinations. Please note. the significant change in $R_2$ was two orders of magnitudes higher than the non-significant change in $R_1$.

## Relaxation measurements in Phantoms

**Accuracy.** $R_1$ and $R_2$ measured using the 3D-QALAS sequence were compared with gold standard measurements (IR for $R_1$ and ME for $R_2$) in the phantoms, see Fig 6. $R_1$ and $R_2$ were significantly larger using QALAS in all fantoms, and the

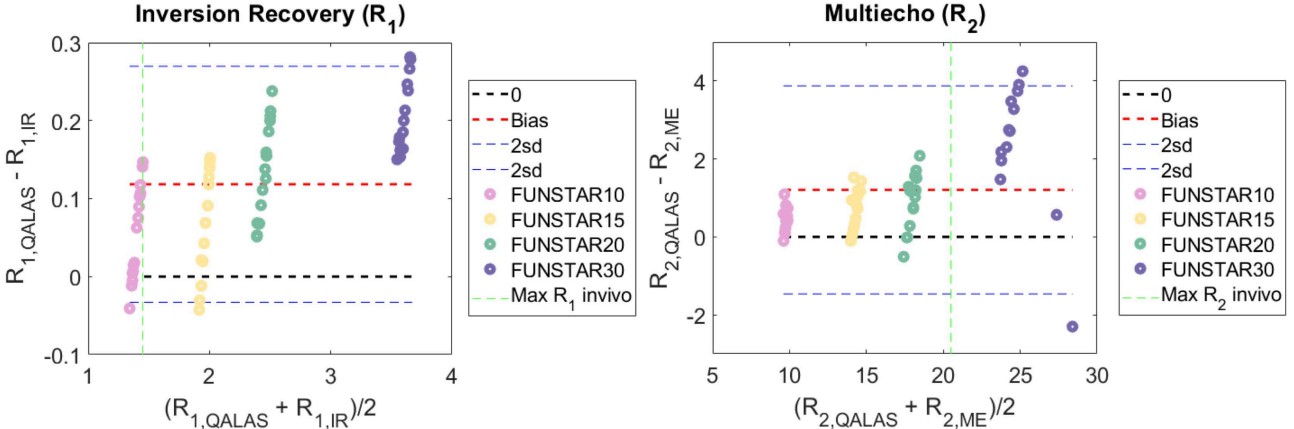

**Fig 6. Phantom measurements.** Quality control and quantification using four different relaxation phantoms (10, 15, 20 and 30 mM). The accuracy of the R1 and R2 measurements using 3D-QALAS sequences compared to gold standard measurements (of R1 and R2) showed a slight difference for R2 and a small difference but only for R1 in the 30 mM phantom. The relaxation values were determined at 15 different positions in the phantoms. Green line shows the maximum value found in vivo.

differences in $R_1$ were 0.06, 0.07, 0.13 0.21 s$^{-1}$ respectively, for the four phantoms (p<0.01). Corresponding differences for $R_2$ were; 0.46, 0.70, 0.93, 2.72 s$^{-1}$ (p<0.001). Note that $R_1$ and $R_2$ were much higher in the 20 mM and 30 mM phantoms than typical physiological values.

**$B_1^+$ inhomogeneities.** The different positions of the phantoms inside the 20-channel head/neck coil, and with the flex coil set up in combination with the head fixation, are shown in S1 and S2 Figs. $B_1^+$ inhomogeneities were also determined in the different positions *within* the phantoms.

When each phantom and phantom position was examined separately, the 20-channel coil and the flex coil measurements displayed slightly linear relations between $R_1$ and $R_2$ (and $B_1^+$) (S1 and S2 Figs). Moreover, when the phantoms were placed at an extreme off-centre position (10 cm of isocentre), the measured $R_1$ and $R_2$ values were different for the anterior ROIs that were near the maximum FOV ('Field of View') for the scanner, compared to more normal positions in the bore. When the phantom was placed 10 cm from isocentre, in 'Position 3', the intra-phantom voxels that were 15 cm from the isocentre began to deviate.

When the centre of each phantoms was positioned 10 cm away from the isocentre, both $R_1$ and $R_2$ values deviated from baseline. This change was primarily due to the inclusion of the most anterior ROIs in the analysis, which were located more than 18 cm from the isocentre (S1 Fig). Notably, this effect disappeared when the anterior ROIs were excluded, as shown in S2 Fig.

## Discussion

The main finding of this study was the observation of a significant difference in $R_2$ in the preoperative compared to the intraoperative setting, in patients. In contrast, no such difference was observed for $R_1$. It was also notable that the increase in $R_2$ was only associated with the *in vivo* measurements and was not observed in the corresponding control experiments in phantoms. This suggests that physiological conditions during surgery probably influenced the transverse relaxation rate of the brain tissue. What those conditions might be is discussed in more detail in the subsequent sections.

### Rapid quantitative 3D relaxometry

In the clinical setting, qualitative radiological assessment is conventionally used rather than relating images to absolute quantities. Nevertheless, it is possible to quantify MR signals and to relate them to physical properties. However, this

technique has previously been too time-consuming, particularly as long scan times would be a major problem in a clinical setting. This issue is further compounded when the technique is used for regular follow-up of young children. Intraoperative MRI has gained increasing importance, attracting growing attention in the field of paediatric brain tumours. However, to our knowledge, there is no reported experience with quantitative relaxometry intraoperatively. To explore relaxometry in the intraoperative environment, the 3D-QALAS method and usability of quantitative values pre – and intraoperatively were studied in the clinical setting.

Previous reviews of using relaxometry for brain tumour diagnostics have emphasised the problem in the clinical setting of long acquisition times associated with quantitative $T_1$ and $T_2$ measurements. Reducing acquisition scan times using qMRI might facilitate diagnosis and treatment selection. Another aspect to consider is the subjective assessment of signal intensities that vary with different tissues, different coils, sequences, or type of scanner. Previous studies regarding contrast measurement with qMRI were performed using a two-dimensional pulse sequence ('*Multidimensional Multi Echo*', MDME, or QRAPMASTER [11,21]) with slice thicknesses of several millimetres. For that reason, partial volume effects in the tumour arise due to mixed tissues in individual voxels [13,22].

3D-QALAS, on the other hand, has been proposed by Kvernby *et al*. [16] as a modality, which could better address this issue and better consider the actual geometry of a tumour. The 3D approach has also been evaluated in neuroradiology by Warntjes *et al.* [17], but not in terms of intraoperative diagnostics. The pulse sequence 3D-QALAS might therefore be better suited because the isotropic 3D acquisition allows for planar reformatting as needed. Moreover, it is optimised compared to other alternatives such as MDME and is therefore even more effective in this situation.

The qMRI technique nevertheless is highly promising. For example, the limited time available for intraoperative MR examinations (typically up to one hour), makes 3D-QALAS an interesting alternative in this setting as proposed by Vargas *et al.* [22]. Also, the fact that extensive information is collected in the same session of approximately 6 min and the possibility to perform segmentation and determining the volume of tissue are valuable features.

## Factors affecting $R_1$ and $R_2$

Relaxometry has previously not been investigated in the interoperative setting, and notably not in children. This study investigates if $R_1$ and $R_2$ depend on $B_1^+$ and/or $B_0$. The results show that the $R_2$ values were higher in the intraoperative measurements in the brain parenchyma, *i.e.,* NAWM and in the thalamus. This was not the case for cerebrospinal fluid.

The key findings for in vivo examinations in this study on the feasibility of using 3D quantitative magnetic resonance imaging by using 3D-QALAS during intraoperative surgery for paediatric brain tumours can be summarised as follows:

1. **$R_2$**: There was a significant $R_2$ difference in brain tissue between intraoperative and preoperative settings, with an $R_2$ increase during intraoperative examinations.

2. **$R_1$**: In contrast, no significant difference was observed in $R_1$ comparing intraoperative and preoperative measurements.

There may be several different explanations for the difference that was observed. Moreover, the specific conditions that the patient experience in the intraoperative setting, as opposed to the pre-operative setting, may affect the relaxometric conditions. No such experiments were performed, neither in any previous nor in this work. In the following paragraphs, we explain the potential chemical and physical phenomena behind these observations. These include chemical effectors, such as osmotic agents, or anaesthesia, geometric factors, and brain structure anisotropy.

In general, water molecules diffuse into (or out of) the cells via osmosis, and the diffusion effects are based on the characteristics and volumes of both intracellular and extracellular fluids. General anaesthesia may temporarily lead to altered white matter microstructure, integrity and volume as well as regional differences [23].

Altered distribution between the intracellular and extracellular space as an effect of medication such as osmotic agents including mannitol, or cortisone, could affect the measured $R_2$ values in MRI performed intraoperatively. For instance, if mannitol is given at surgery to reduce intracranial pressure associated with cerebral oedema [24], this may also influence

the brain parenchyma, and there may well be a $T_2$ effect consistent with what has been observed previously [25]. The effect of anaesthetic drugs on intracranial pressure and cerebral blood flow, or metabolism and structural changes at the molecular level also need to be evaluated in this context.

An earlier study found that general anaesthesia was associated with a significant increase of $T_2$ [25], but only in specific brain regions (the pons and corpus callosum). In contrast, $T_1$ values were not significantly affected. This suggests that increased $T_2$ may be associated with structural changes in the brain at a molecular level, thereby affecting the aqueous environment and therefore the relaxation conditions of water.

Observations using diffusion tensor imaging on healthy adults exposed to sevoflurane anaesthesia caused (transient) microstructural changes in the brain tissue [23]. Volumetric changes of the brain tissue have been observed, including a decrease in white matter volume, an increase in grey matter volume as well as of cerebrospinal fluid during anaesthesia. There are several proposed explanations for these observations, including the shrinking of glial cells (due to decreased neural activity, similar to what is expected to happen during sleep). Such shrinking could also explain the observed increase of the extracellular space in the tissue.

There has also been a proposal that the glymphatic system is affected by anaesthesia, an effect that has been observed in animal studies [26]. Clearly, it would be important to consider the structural effects of anaesthesia when examining patients by MRI, especially when comparing data with preoperatively acquired images without anaesthesia.

However, the effects of anaesthesia are not exclusively limited to changes in microstructure. A body temperature decrease as a consequence of the impairment of thermoregulation by anaesthesia has also been observed in paediatric patients [27], although this is compensated to a slight degree by the RF-induced heating during an MR examination, independent of field strength, which may affect paediatric patients more than adults [28]. A lowered temperature will affect water diffusion and therefore also relaxation, at least for $T_1$, though not so much for $T_2$ [29]. In summary, the general reported observations tend to suggest that $T_2$ will increase because of exposure to anaesthesia [23,25,27].

## Head and coil positioning

The head of the patient is fixed in a headrest, with limited options to place coils in an optimal position. There is therefore a substantial risk that it might not be possible to position the patient suitably in the bore of the MR scanner, with respect to the isocentre. According to a recent study describing intraoperative MRI [30], the acquisition of MR images in the prone position was shown to benefit from a smaller sensitivity to artefacts. In addition, the SNR increased when four instead of two detection coils were used [30]. In our work, a 20–channel or a 64-channel head/neck coil was used preoperatively, whereas a flex coil was positioned in the head fixation underneath the patient's head, and another smaller flex coil was placed on top of the patient's head, after draping (see Fig 2). However, although it affects the available SNR, one would not expect an effect on water relaxation because of the use of different detection coils.

## Fibre tract orientation

Craniotomy per se is a physiological factor that can influence the measured values as well as both brain shift and head orientation, *i.e.,* fibre tract orientation, as the parenchyma often anatomically shifts during the operation. White matter is a highly anisotropic tissue and the $T_2$ signal decay in white matter is intrinsically also multi-exponential [31]. The $T_2$ relaxation is substantially shorter for components representing myelin-associated water. In contrast, relaxation components representing the pools of both intra-axonal and extra-axonal water are characterised by longer $T_2$ [32].

Tilting of the head has also been shown to affect $R_2$ and the difference from baseline value depends on the direction in which the neural fibre tracts are tilted with respect to the main magnetic field ($B_0$) [32]. In other words, the extent to which the head of the patient is bent, in which direction, and around which cardinal axis will affect $R_2$ [32,33]. The relaxation values for both myelin-associated water and other tissue water pools are influenced by the anisotropic structure as this also closely depends on the angulation of the fibre bundles with respect to the direction of the main magnetic field ($B_0$) of the MR scanner.

The longest $T_2$ times have previously typically been observed when the fibre direction was parallel to $B_0$, whereas the shortest were observed orthogonally to the static magnetic field. The effects of anisotropy have been found to affect extracellular water the most, whereas intra-axonal water was affected the least. The orientation effects have been attributed to a combination of dipole-dipole interactions and susceptibility effects [32–35].

In summary, the generally reported observations tend to suggest that $T_2$ will change as an effect of an orientation that is off the static-field axis.

## Treatment of cerebral oedema

During neurosurgery, conditions such as cerebral oedema and elevated intracranial pressure may occur. These can be treated with osmotic agents, and mannitol is one such agent that is widely used [24]. Mannitol is a carbohydrate polyol containing six carbons, each with a hydroxy group attached, and therefore it interacts strongly with the aqueous environment [36]. Typical concentrations used for mannitol infusion for clinical purposes are in the order of 20% (w/v), which is relatively close to the solubility of mannitol. Such a high concentration of mannitol is very hypertonic, and it corresponds to about 3% (w/v) hypertonic saline (cf. isotonic saline which is 0.9%). A 20% (w/v) mannitol solution corresponds to a molar concentration of about 1 M, which is quite substantial [37].

The osmotic agent reduces brain oedema by shifting water from the intracellular environment to the extracellular space, *provided* that the blood-brain barrier is intact and not perturbed in any manner [24]. This barrier is impermeable to mannitol. Mannitol therefore acts as an osmotic agent, but it also affects the viscosity of the extracellular space [38]. It has been reported that the addition of mannitol decreases the viscosity of whole blood [24], which in turn increases oxygen delivery to the brain. It is not entirely clear how the compounded effects of such conditions would affect the $T_2$ of the brain tissue.

To the best of our knowledge, there are no previous reports on the relaxation effects of mannitol (or other osmotic agents) *in vivo*. Nevertheless, both changed viscosity and enlarged extracellular aqueous pools would clearly affect the relaxation of extracellular water [24,36–39]. However, no experimentation in this study specifically tested for these effects in patient brains, and therefore it remains to be done using a suitable experimental design.

*In Summary:* In this work we observed a significantly increased $R_2$ (corresponding to a decreased $T_2$) for several NAWM and thalamus locations in the paediatric brain during intraoperative surgery. As the setting typically involves an off-centre position of the brain, the application of anaesthesia and the use of an osmotic agent (mannitol), the effect of relaxation is clearly multifactorial. Nevertheless, we believe that the main contributors to the observed reduced $T_2$ are the comparatively different angulations of the brain, as well as the use of a high concentration of an osmotic agent during surgery. Although anaesthesia may also contribute to the observed change in relaxation, it would be in the opposite direction and therefore less likely to influence the measurements.

## Strengths and limitations

This work has some limitations, as well as strengths. We have not included any adult patients for comparison, something that would have contributed to a better understanding of the observations of NAWM and the thalamus.

In addition, the intraoperative environment is very challenging from several perspectives, which complicates both evaluation and conclusions. One such challenge is that intraoperative qMRI requires the use of less sensitive and spatially controlled flex coils, instead of a conventional head coil which offers higher SNR and the capability to perform accelerated acquisition. Also, the head is not placed in the isocentre of the magnet which means the $B_0$, (especially when the scanner's baseline shim settings are used) and $B_1^+$ field will be less optimal with respect to the magnet design. It might be advantageous to perform additional technical feasibility studies of the entire setting, in comparison with conventional preoperative setups using head coils.

To adjust for the effect of the distance between the flex coils and isocentre between each examination, we used custom-designed *in vitro* relaxation phantoms. However, a limitation of this approach is that phantoms are isotropic while

brain tissues are anisotropic, and it is therefore not possible to make direct comparisons. Moreover, based on the nature of the study, which entailed intraoperative paediatric examinations, it was not possible to compare with healthy control subjects with respect to all tissue properties *in vivo*.

In the surgical setting, some ROIs were located 15 cm or more from the magnet isocentre. *In vitro*, this led to banding artifacts in the anterior part of the phantom in the magnitude images of 3D-QALAS acquisitions, particularly after the first and second readouts following the T2-prep and inversion pulses (see Fig 3A).

No banding was observed neither in the fourth or fifth readouts, but phase wraps appeared in the phase images, indicating certain $B_0$ inhomogeneity—likely due to the absence of dedicated large field-of-view $B_0$ shimming. Future studies should include additional $B_0$ shimming, especially when imaging away from isocentre.

Additional errors stemmed from sequence design, such as imperfect magnetization after T2-prep and suboptimal inversion pulses. These could be improved by increasing the number of BIREF-1 pulses, as shown in [18]. Minor $B_1^+$ variation across the FOV explained only a small portion of the $R_2$ changes. In extreme phantom positions, $B_1^+$ map quality was mainly degraded by strong local $B_0$ inhomogeneities, as noted above.

In this report we aimed to introduce a proof-of-concept and assess the comparability of relaxometry measurements in MR examinations performed both pre- and intraoperatively in healthy tissue, during anaesthesia. Some data were missing, for example, with some patients contrast agent was only given preoperatively.

Finally, it is important to mention that there are also age-related aspects to consider when studying young children, such as differences in the degree of myelination at different ages and development. Fortunately, in paediatric oncology in general, the cohort is small, but it is also heterogeneous with a wide range of different tumour types. In addition, determining the myelin concentrations were outside the scope of the present work.

In summary, there was a significant change in $R_2$ *in vivo* that was not observed in the control phantom measurements. The main strengths of the study design are its prospective design, and the carefully performed control experiments using phantoms. With respect to the *in vivo* measurements, we used paired intra-individual tests both with and without the administration of a contrast agent.

## Conclusion

This study shows that $R_1$ and $R_2$ values can be measured quantitatively in the intraoperative setting using 3D-QALAS and flex coils, with only a slight bias due to variation in $B_1^+$. The $R_2$ values of the brain parenchyma were higher during surgery. These findings suggest that while $R_1$ measurements remain consistent, $R_2$ measurements are affected by the intraoperative setting, due to physiological conditions during surgery. This needs to be considered when performing relaxometry in the intraoperative setting.

## Supporting information

**S1 Fig. Illustration of the phantom control measurements.** (A) The measurements were performed in four different configurations. The left panel shows one of the four phantoms (10, 25, 20 and 30 mM) centred within a head/neck coil (coil in grey, and phantom in red). The three setups to the right are with a phantom and the two flex coils positioned at 0 cm from the centre, at 5 cm from the centre and 10 cm from the centre, respectively. The phantom was placed in the head fixation. (B) $R_1$ and $R_2$ (s$^{-1}$) respectively, as a function of $B_1^+$ (in a.u., where 1.0 represents an expected $B_1^+$ vs flip angle relation). Four different phantoms were used, placed in different positions with respect to the isocentre. The ROI positions within each phantom varied from A70 to P70. $B_1^+$ were typically in the range 0.9 to 1.1, except in the more extreme phantom positions. (C) R1 and R2 (s$^{-1}$) respectively, as a function of bore position of the ROIs. The ROI positions within each phantom varied from A70 to P70. N.B. The cases when the phantoms and ROIs were placed extremely far off centre (although within the max FOV of the scanner) resulted in large offsets.
(TIFF)

**S2 Fig. Phantom control measurements, excluding anterior ROIs.** (A) The measurements were performed in four different configurations. The left panel shows one of the four phantoms (10, 25, 20 and 30 mM) centred within a head/neck coil (coil in grey, and phantom in red). The three setups to the right are with a phantom and the two flex coils positioned at 0 cm from the centre, at 5 cm from the centre and 10 cm from the centre, respectively. The phantom was placed in the head fixation. (B) $R_1$ and $R_2$ (s$^{-1}$) respectively, as a function of $B_1^+$ (in a.u., where 1.0 represents an expected $B_1^+$ vs flip angle relation). Four different phantoms were used, placed in different positions with respect to the isocentre. The ROI positions within each phantom varied from centre to P70. $B_1^+$ were typically in the range 0.9 to 1.1. (C) $R_1$ and $R_2$ (s$^{-1}$) respectively, as a function of bore position of the ROIs. The ROI positions within each phantom varied from centre to P70. N.B. Excluding anterior ROIs only small variation are observed.
(TIFF)

**S3 Fig. Schematic overview of the positioning of ROIs in the phantoms.** All ROI placements in the AP direction in the phantoms. The images are shown in axial view.
(TIFF)

**S1 Data. Complete protocol parameters.**
(PDF)

**S2 Data. CodefitROI2step.m.** Code for fitting QALAS data.
(M)

**S3 Data. CodemappT2.m.** Code for Gold standard T2-mapping.
(M)

**S4 Data. CodemodelQALAS.m.** Subfunction used by CodefitROI2step.
(M)

## Acknowledgments

Research engineers Jens Tellman, and Mary Adjeiwaah, are gratefully acknowledged for their skilled contribution to this work.

## Author contributions

**Conceptualization:** Per Nyman, Ida Blystad, Peter Lundberg, Anders Tisell.

**Data curation:** Anders Tisell.

**Formal analysis:** Per Nyman, Ida Blystad, Anders Tisell.

**Funding acquisition:** Ida Blystad, Peter Lundberg.

**Investigation:** Per Nyman, Rafael Turczynski Holmgren, Anna Ljusberg, Frederik Testud, Ida Blystad, Anders Tisell.

**Methodology:** Per Nyman, Anders Tisell.

**Project administration:** Emma Nordh, Peter Lundberg.

**Software:** Frederik Testud, Anders Tisell.

**Writing – original draft:** Per Nyman, Frederik Testud, Ida Blystad, Peter Lundberg, Anders Tisell.

**Writing – review & editing:** Per Nyman, Rafael Turczynski Holmgren, Emma Nordh, Anna Ljusberg, Oscar Snödahl, Frederik Testud, Ida Blystad, Peter Lundberg, Anders Tisell.

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
