## [Decision Letter · Decision Letter 0]

6 Oct 2025

Dear Dr. NYMAN,

Thank you for submitting your manuscript to PLOS ONE. After careful consideration, we feel that it has merit but does not fully meet PLOS ONE’s publication criteria as it currently stands. Therefore, we invite you to submit a revised version of the manuscript that addresses the points raised during the review process.

We look forward to receiving your revised manuscript.

Kind regards,

Jochen Leupold

Academic Editor

PLOS ONE

Journal Requirements:

“I have read the journal's policy and the author Frederik Testud of this manuscript have the following competing interests: Employee of Siemens Healthineers AB, Sweden”

We note that one or more of the authors are employed by a commercial company: Siemens Healthineers AB, Sweden

5. We note that you have indicated that there are restrictions to data sharing for this study. PLOS only allows data to be available upon request if there are legal or ethical restrictions on sharing data publicly. For more information on unacceptable data access restrictions, please see http://journals.plos.org/plosone/s/data-availability#loc-unacceptable-data-access-restrictions.

7. Please include a copy of Table 1 which you refer to in your text on page 4.

Additional Editor Comments:

Dear Authors,

as you can see, both reviewers recommend a major revision of the manuscript. So please carefully consider the reviewers’ comments and modify the manuscript accordingly or provide a well-founded rebuttal if you disagree.

Best regards,

Jochen Leupold

Reviewers' comments:

Reviewer's Responses to Questions

**Comments to the Author**

1. Is the manuscript technically sound, and do the data support the conclusions?

Reviewer #1: Yes

Reviewer #2: Partly

2. Has the statistical analysis been performed appropriately and rigorously?

Reviewer #1: I Don't Know

Reviewer #2: I Don't Know

3. Have the authors made all data underlying the findings in their manuscript fully available?

Reviewer #1: No

Reviewer #2: No

4. Is the manuscript presented in an intelligible fashion and written in standard English?

Reviewer #1: Yes

Reviewer #2: Yes

Reviewer #1: In "Intraoperative 3D quantitative magnetic resonance imaging in paediatric brain tumour surgery" the authors investigate in both phantoms and surgical patients the change in R1 and R2 between preoperative and interoperative qMRI. The authors find that in phantoms and in vivo, R1 is consistent between pre- and interoperative scanning, but R2 increases in vivo. Because phantom R2 is the same between both scanning situations, the authors attribute change in R2 to physiologic changes during surgery.

Although the work is very good, I have to recommend major revision, as Table 1 was not included in the manuscript, and thus could not be reviewed.

Generally, figures should be numbered in order of their appearance. The present order of first mention of figures appears to be 1, 2, 3, 7, 6, 4, 5,

P4, Magnetic Resonance, Data acquisition section, paragraph 1: Figure 1 does not contain what it is stated to contain in this paragraph. Perhaps this is meant to refer to what is presently Figure 3?

P5, Patient MR Measurements, paragraph 4: Table 1 cannot be found.

Reviewer #2: In this study, the authors measured R1 and R2 relaxation values using 3D QALAS MRI to evaluate it as an alternative to the gold-standard sequences that typically require longer duration. Compared with the preoperative head coil, they used flex coils for intraoperative scanning and compared the results. By placing the ROIs at various locations in the brain, they examined whether B1+ inhomogeneities also affect the R1 and R2 values. Although it has significant limitations, this study is a valuable proof-of-concept step towards using quantitative MRI parameters to assist in operative procedures.

Major comments:

- In relation to the gadolinium administration, it is unclear whether the authors observed any differences in R1. Since they mentioned scanning both before and after Gd contrast agent administration, there is no discussion on how this may have affected R1. Although R1 may not be significant, the statement in the patients examinations section in the results “R2 was significantly higher in the intraoperative compared to preoperative examinations, both with and without contrast agent” seems confusing, since Gadolinium is an R1 contrast agent and generally has less impact on R2. Which specific Gd-based contrast agent was used? The authors also did not include any information about it in the discussion.

- For the contrast agent studies, how was the B1+ difference calculated? What is the significance of -0.041 s-1 with no gadolinium contrast agent, and 0.047 s-1 with gadolinium contrast agent? The authors need to elaborate on this in the discussion.

- Although the authors discussed it as a limitation of the current study, how much time would increase if the authors had to perform active shimming? At least for the isotropic phantom samples, this comparison should be carried out to determine whether adding an active shimming step before scanning enhances the B1+ inhomogeneity and whether the R1 and R2 values under that condition are consistent with those calculated from the gold standard experiments, particularly in areas farther from the centre.

- The authors highlight a significant difference in R2 pre-operatively and intraoperatively as the main finding. However, apart from the absolute values, it is essential to specify the percentage difference observed and clarify what constitutes a 'significant difference,' given that the R2 values are at least one order of magnitude higher than R1. The authors should also elaborate on this aspect. While I agree the manuscript would benefit from the authors’ reflections on why a significant change in R2 was observed in vivo, the current emphasis on anaesthesia, mannitol, and fibre tract orientation seems too hypothetical, especially in the absence of any tests done for these factors.

- In the section on the treatment of cerebral oedema, the authors mention “there are no previous reports on the relaxation effects of mannitol (or other osmotic agents) in vivo." I think they need to clarify that no experimentation in this study specifically tests for these effects. From a discussion perspective, I believe the section on anaesthesia and mannitol can be combined, since the detailed discussion on them appears to be the authors proposing explanations based on the experiments shown in this study, which is not the case.

- Based on the experiments conducted in this study, how do the authors envisage the calculation of R1 and R2 using 3D QALAS being applied to determining tumour volume for resection or estimating myelin concentration? This explanation should be included in the conclusion, as the current statement is too vague for a quantitative study.

- Figure 2 caption: “The 3D-QALAS sequence consists of five dynamics.” Dynamics of what? The sentence is unclear. Also, it is not mentioned what the blue-to-orange scale represents. Assuming it is signal intensity, can the range be made the same for pre-operative and intraoperative measurements for better comparison?

- Figure 4a. Regarding the R1 and R2 correlation, are any specific clustering patterns observed, such as based on distance from the magnet centre for particular brain regions, according to specific ROIs? Currently, it is unclear whether the ROIs shown on the left are similar regions in different OP conditions.

- Figure 5. From the figure, it is unclear whether the average bias is the same for all four phantoms with different concentrations, or if the average bias is based on a specific concentration or patient data, with the phantom values overlaid to show the pattern.

- Any comments and citing the references regarding the expected B1+ in relation to flip angle would benefit the manuscript, since this factor has been used in the experiments and results in Figure 6.

- Labelling the A70 to P70 positions on an MRI scan and including them in a supplementary figure will be helpful, since the current representative figure 7 does not show the positions of all 15 ROIs.

Minor comments:

- Clarify what differences were observed between the head coil and flex coil. Were those restricted to just SNR differences?

- The authors mentioned that the R1 and R2 values can assist in tissue classification and estimating myelin concentration both in the introduction and discussion. Based on the experiments conducted, did the authors attempt to calculate or estimate these values for the 10 patients? If yes, what were the results? If no, what are the limitations? If this is outside the scope of the study, they should simply mention it in the discussion.

- For the in-house MATLAB code, a reference or citation to previous reports or a GitHub link should be provided.

- While the phantoms are isotropic and the authors pointed out this limitation in the discussion, they should also suggest better alternatives for future experiments in this study. Not only should the anisotropic nature of the brain be considered, but also the presence of R2-impacting ions like iron and zinc.

- The discussion on anaesthesia affecting the increase of T2 is interesting; however, there are no experiments in this study conducted to test or support this claim. Therefore, a dedicated discussion on this point seems somewhat of a stretch, especially since the anisotropy of brain tissue structure and mineral content with ions affecting R1 and R2 are not discussed at all. Similarly, unless the authors tested for fibre tract orientation or analysed their data to be relevant, the discussion does not seem applicable to the findings.

- What do the authors interpret as the reason for the difference in their findings being due to the prone position in the intraoperative setting?

- The detail in the data analysis part “The ROIs were placed by an experienced (15+ years) neuroradiologist (IB) on” would be better suited to the acknowledgements or contribution section.

The study is a promising proof-of-concept, but the manuscript's writing is disjointed. Refining the purpose, experiments, and discussion will enhance clarity. Basic formatting improvements are recommended, such as adding full stops where they are missing in the introduction and methods sections.

**Do you want your identity to be public for this peer review?** For information about this choice, including consent withdrawal, please see our Privacy Policy

Reviewer #1: No

Reviewer #2: No

---

## [Author Response · Author response to Decision Letter 1]

30 Nov 2025

Please see file 'Response to reviewers IntraOP_QALAS_PlosOne'

---

## [Decision Letter · Decision Letter 1]

18 Jan 2026

Dear Dr. NYMAN,

We look forward to receiving your revised manuscript.

Kind regards,

Jochen Leupold

Academic Editor

PLOS One

Journal Requirements:

Additional Editor Comments:

Dear Authors,

please apologise the delay that was caused by the non-response of reviewer 1. However, since you have addressed the comments of that reviewer, it is not necessary to wait any longer.

Also, you have addressed the comments of reviewer 2 and upon her reccomendation your manuscript is accepted. However, since there are a few language issues raised by reviewer 2, the official decision is "minor revision". Note this is only for technical reasons, so please quickly provide a manuscript with only these language issues corrected, which I will accept immediately upon receipt. This is probably cleaner than doing these corrections only in the proof.

Sincerely,

Jochen Leupold

Reviewers' comments:

Reviewer's Responses to Questions

**Comments to the Author**

Reviewer #2: All comments have been addressed

2. Is the manuscript technically sound, and do the data support the conclusions?

Reviewer #2: Yes

3. Has the statistical analysis been performed appropriately and rigorously?

Reviewer #2: I Don't Know

4. Have the authors made all data underlying the findings in their manuscript fully available?

Reviewer #2: Yes

5. Is the manuscript presented in an intelligible fashion and written in standard English?

Reviewer #2: Yes

Reviewer #2: The authors have addressed my concerns and taken my suggestions into account to my satisfaction. I appreciate the potential of this study in intraoperative diagnostic imaging in paediatric brain surgery.

There are minor language suggestions that the authors can consider enhancing the clarity of the manuscript, which they may address at their discretion.

1. Abstract : Last line under the conclusion heading, change “R2 tissue values” to “R2 values in tissue”.

2. Introduction: The sentence “Concerning paediatric brain tumours there is little or no knowledge concerning the use of qMRI and to our knowledge none in the perioperative setting during paediatric brain tumour surgery”, could be rephrased to “Concerning pediatric brain tumors, there is little to no knowledge about the use of qMRI, and, to our knowledge, none in the perioperative setting during pediatric brain tumor surgery” for clarity.

“And also that R1 and R2 measurements were affected by different receive coils. ” should be rephrased to “Additionally, it was examined whether the R1 and R2 measurements were affected by different receiver coils”.

3. Results: The previous section title “Patient Examination” was better than “patient”. Last line in this section “N.B.” can be changed to “Please note”. The previous section title “Relaxation measurement in phantoms” was better than “phantoms”.

4. Discussion: Under the factors affecting R1 and R2 section, the sentence “In the following we therefore attempt to explore which factors, whether chemical of physical, that potentially would explain the observations.” A word seems to be missing after ‘following’. The sentence could be replaced with “In the following paragraphs, we explain the potential chemical and physical phenomena behind these observations.”

Since there is only one sub-heading after this paragraph, either the title “chemical effectors and structure of brain tissue” could be removed, or an appropriate heading like “Anaesthesia” could be given to the context in the following paragraphs.

**Do you want your identity to be public for this peer review?** For information about this choice, including consent withdrawal, please see our Privacy Policy

Reviewer #2: **Yes:** Bhargy Sharma

---

## [Author Response · Author response to Decision Letter 2]

21 Jan 2026

All remaining language issues were corrected.

/PN

---

## [Editor Report · Decision Letter 2]

22 Jan 2026

Intraoperative 3D quantitative magnetic resonance imaging in paediatric brain tumour surgery

PONE-D-25-46877R2

Dear Dr. NYMAN,

We’re pleased to inform you that your manuscript has been judged scientifically suitable for publication and will be formally accepted for publication once it meets all outstanding technical requirements.

Kind regards,

Jochen Leupold

Academic Editor

PLOS One

Additional Editor Comments (optional):

Dear Authors,

the few remaining issues have been addressed, so the manuscript is now accepted. Congratulations!

Sincerely,

Jochen Leupold

---

## [Editor Report · Acceptance letter]

PONE-D-25-46877R2

PLOS One

Dear Dr. NYMAN,

I'm pleased to inform you that your manuscript has been deemed suitable for publication in PLOS One. Congratulations! Your manuscript is now being handed over to our production team.

Kind regards,

on behalf of

Dr. Jochen Leupold

Academic Editor

PLOS One